# Mutual Information Gradient Estimation for Representation Learning

**Liangjian Wen**[1,2]**, Yiji Zhou**[1]**, Lirong He**[1]**, Mingyuan Zhou**[3]**, Zenglin Xu**[4,2,1]
[1] SMILE Lab, School of Computer Science and Engineering
University of Electronic Science and Technology of China, Chengdu, China
[2] Center for Artificial Intelligence
Peng Cheng Laboratory, Shenzhen, China
[3] McCombs School of Business
University of Texas at Austin, Austin, United States
[4] School of Computer Science and Technology
Harbin Institute of Technology, Shenzhen, China
`wlj6816@gmail.com,zhouyiji@outlook.com,ronghe1217@gmail.com,`
`mingyuan.zhou@mccombs.utexas.edu,xuzenglin@hit.edu.cn`

## Abstract

Mutual Information (MI) plays an important role in representation learning. However, MI is unfortunately intractable in continuous and high-dimensional settings. Recent advances establish tractable and scalable MI estimators to discover useful representation. However, most of the existing methods are not capable of providing an accurate estimation of MI with low-variance when the MI is large. We argue that directly estimating the gradients of MI is more appealing for representation learning than estimating MI in itself. To this end, we propose the Mutual Information Gradient Estimator (MIGE) for representation learning based on the score estimation of implicit distributions. MIGE exhibits a tight and smooth gradient estimation of MI in the high-dimensional and large-MI settings. We expand the applications of MIGE in both unsupervised learning of deep representations based on InfoMax and the Information Bottleneck method. Experimental results have indicated significant performance improvement in learning useful representation.

## 1 Introduction

Mutual information (MI) is an appealing metric widely used in information theory and machine learning to quantify the amount of shared information between a pair of random variables. Specifically, given a pair of random variables $\mathbf{x}, \mathbf{y}$, the MI, denoted by $I(\mathbf{x}; \mathbf{y})$, is defined as

$$I(\mathbf{x}; \mathbf{y}) = \mathbb{E}_{p(\mathbf{x},\mathbf{y})} \left[ \log \frac{p(\mathbf{x}, \mathbf{y})}{p(\mathbf{x})p(\mathbf{y})} \right], \tag{1}$$

where $\mathbb{E}$ is the expectation over the given distribution. Since MI is invariant to invertible and smooth transformations, it can capture non-linear statistical dependencies between variables (Kinney & Atwal, 2014). These appealing properties make it act as a fundamental measure of true dependence. Therefore, MI has found applications in a wide range of machine learning tasks, including feature selection (Kwak & Choi, 2002; Fleuret, 2004; Peng et al., 2005), clustering (Müller et al., 2012; Ver Steeg & Galstyan, 2015), and causality (Butte & Kohane, 1999). It has also been pervasively used in science, such as biomedical sciences (Maes et al., 1997), computational biology (Krishnaswamy et al., 2014), and computational neuroscience (Palmer et al., 2015).

Recently, there has been a revival of methods in unsupervised representation learning based on MI. A seminal work is the InfoMax principle (Linsker, 1988), where given an input instance $x$, the goal of the InfoMax principle is to learn a representation $E_\psi(x)$ by maximizing the MI between the input and its representation. A growing set of recent works have demonstrated promising empirical performance in unsupervised representation learning via MI maximization (Krause et al., 2010; Hu

et al., 2017; Alemi et al., 2018b; Oord et al., 2018; Hjelm et al., 2019). Another closely related work is the Information Bottleneck method (Tishby et al., 2000; Alemi et al., 2017), where MI is used to limit the contents of representations. Specifically, the representations are learned by extracting task-related information from the original data while being constrained to discard parts that are irrelevant to the task. Several recent works have also suggested that by controlling the amount of information between learned representations and the original data, one can tune desired characteristics of trained models such as generalization error (Tishby & Zaslavsky, 2015; Vera et al., 2018), robustness (Alemi et al., 2017), and detection of out-of-distribution data (Alemi et al., 2018a).

Despite playing a pivotal role across a variety of domains, MI is notoriously intractable. Exact computation is only tractable for discrete variables, or for a limited family of problems where the probability distributions are known. For more general problems, MI is challenging to analytically compute or estimate from samples. A variety of MI estimators have been developed over the years, including likelihood-ratio estimators (Suzuki et al., 2008), binning (Fraser & Swinney, 1986; Darbellay & Vajda, 1999; Shwartz-Ziv & Tishby, 2017), k-nearest neighbors (Kozachenko & Leonenko, 1987; Kraskov et al., 2004; Pérez-Cruz, 2008; Singh & Póczos, 2016), and kernel density estimators (Moon et al., 1995; Kwak & Choi, 2002; Kandasamy et al., 2015). However, few of these mutual information estimators scale well with dimension and sample size in machine learning problems (Gao et al., 2015).

In order to overcome the intractability of MI in the continuous and high-dimensional settings, Alemi et al. (2017) combines variational bounds of Barber & Agakov (2003) with neural networks for the estimation. However, the tractable density for the approximate distribution is required due to variational approximation. This limits its application to the general-purpose estimation, since the underlying distributions are often unknown. Alternatively, the Mutual Information Neural Estimation (MINE, Belghazi et al. (2018)) and the Jensen-Shannon MI estimator (JSD, Hjelm et al. (2019)) enable differentiable and tractable estimation of MI by training a discriminator to distinguish samples coming from the joint distribution or the product of the marginals. In detail, MINE employs a lower-bound to the MI based on the Donsker-Varadhan representation of the KL-divergence, and JSD follows the formulation of f-GAN KL-divergence. In general, these estimators are often noisy and can lead to unstable training due to their dependence on the discriminator used to estimate the bounds of mutual information. As pointed out by Poole et al. (2019), these unnormalized critic estimators of MI exhibit high variance and are challenging to tune for estimation. An alternative low-variance choice of MI estimator is Information Noise-Contrastive Estimation (InfoNCE, Oord et al. (2018)), which introduces the Noise-Contrastive Estimation with flexible critics parameterized by neural networks as a bound to approximate MI. Nonetheless, its estimation saturates at log of the batch size and suffers from high bias. Despite their modeling power, none of the estimators are capable of providing accurate estimation of MI with low variance when the MI is large and the batch size is small (Poole et al., 2019). As supported by the theoretical findings in McAllester & Statos (2018), any distribution-free high-confidence lower bound on entropy requires a sample size exponential in the size of the bound. More discussions about the bounds of MI and their relationship can be referred to Poole et al. (2019).

In summary, existing estimators first approximate MI and then use these approximations to optimize the associated parameters. For estimating MI based on any finite number of samples, there exists an infinite number of functions, with arbitrarily diverse gradients, that can perfectly approximate the true MI at these samples. However, these approximate functions can lead to unstable training and poor performance in optimization due to gradients discrepancy between approximate estimation and true MI. Estimating gradients of MI rather than estimating MI may be a better approach for MI optimization. To this end, to the best of our knowledge, we firstly propose the Mutual Information Gradient Estimator (MIGE) in representation learning. In detail, we estimate the score function of an implicit distribution, $\nabla_{\mathbf{x}} \log q(\mathbf{x})$, to achieve a general-purpose MI gradient estimation for representation learning. In particular, to deal with high-dimensional inputs, such as text, images and videos, score function estimation via Spectral Stein Gradient Estimator (SSGE) (Shi et al., 2018) is computationally expensive and complex. We thus propose an efficient high-dimensional score function estimator to make SSGE scalable. To this end, we derive a new reparameterization trick for the representation distribution based on the lower-variance reparameterization trick proposed by Roeder et al. (2017).

We summarize the contributions of this paper as follows:

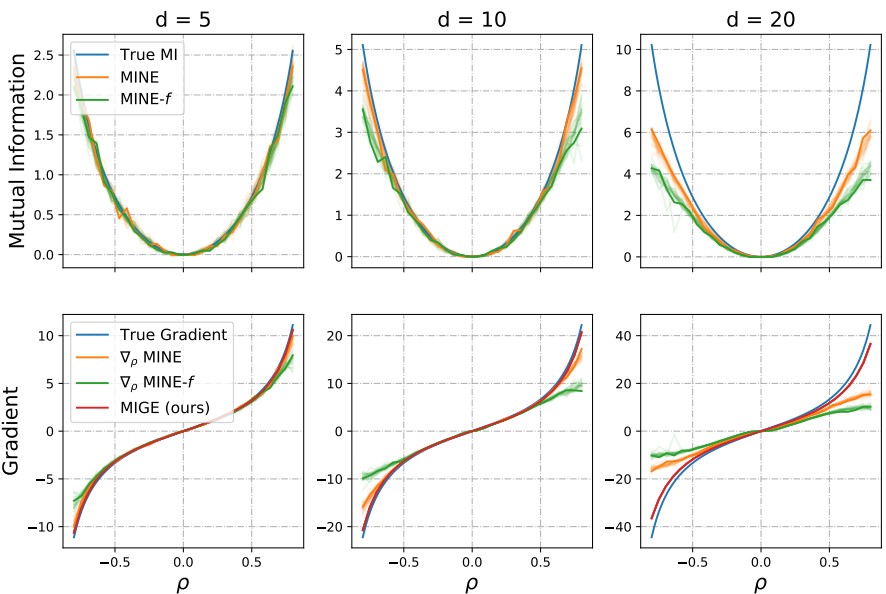

Figure 1: Estimation performance of MINE, MINE-$f$ and MIGE. Each estimation approach has been taken additional 20 times and plotted with light curves. **Top:** True MI and corresponding estimation of MINE and MINE-$f$. **Bottom:** True gradient and corresponding estimation of MINE, MINE-$f$ and MIGE. Our approach MIGE only appears in bottom figures since it directly gives gradient estimation. As we observe, MIGE gives more stable, smooth and accurate results.

- We propose the Mutual Information Gradient Estimator (MIGE) for representation learning based on the score function estimation of implicit distributions. Compared with MINE and MINE-$f$, MIGE provides a tighter and smoother gradient estimation of MI in a high-dimensional and large-MI setting, as shown in Figure 1 of Section 4.
- We propose the Scalable SSGE to alleviate the exorbitant computational cost of SSGE in high-dimensional settings.
- To learn meaningful representations, we apply SSGE as gradient estimators for both InfoMax and Information Bottlenck, and have achieved improved performance than their corresponding competitors.

## 2 SCALABLE SPECTRAL STEIN GRADIENT ESTIMATOR

Score estimation of implicit distributions has been widely explored in the past few years (Song et al., 2019; Li & Turner, 2017; Shi et al., 2018). A promising method of score estimation is the Stein gradient estimator (Li & Turner, 2017; Shi et al., 2018), which is proposed for implicit distributions. It is inspired by generalized Steins identity  (Gorham & Mackey, 2015; Liu & Wang, 2016) as follows.

**Steins identity.** Let $q(\mathbf{x})$ be a continuously differentiable (also called smooth) density supported on $\mathcal{X} \subseteq \mathbb{R}^d$, and $\boldsymbol{h}(\mathbf{x}) = [h_1(\mathbf{x}), h_2(\mathbf{x}), \ldots, h_{d'}(\mathbf{x})]^{\mathrm{T}}$ is a smooth vector function. Further, the boundary conditions on $\boldsymbol{h}$ is

$$q(\mathbf{x})\boldsymbol{h}(\mathbf{x}) = 0, \forall \mathbf{x} \in \partial\mathcal{X} \text{ if } \mathcal{X} \text{ is compact, or } \quad \lim_{\mathbf{x}\to\infty} q(\mathbf{x})\boldsymbol{h}(\mathbf{x}) = 0 \text{ if } \mathcal{X} = \mathbb{R}^d. \tag{2}$$

Under this condition, the following identity can be easily checked using integration by parts, assuming mild zero boundary conditions on $\boldsymbol{h}$,

$$\mathbb{E}_q \left[ \boldsymbol{h}(\mathbf{x})\nabla_{\mathbf{x}} \log q(\mathbf{x})^{\mathrm{T}} + \nabla_{\mathbf{x}}\boldsymbol{h}(\mathbf{x}) \right] = \mathbf{0}. \tag{3}$$

Here $\boldsymbol{h}$ is called the Stein class of $q(\boldsymbol{x})$ if Steins identity Eq. (3) holds. Monte Carlo estimation of the expectation in Eq. (3) builds the connection between $\nabla_{\mathbf{x}} \log q(\mathbf{x})$ and the samples from $q(\mathbf{x})$ in Steins identity. For modeling implicit distributions, Motivated by Steins identity, Shi et al. (2018) proposed Spectral Stein Gradient Estimator (SSGE) for implicit distributions based on Stein's identity and a spectral decomposition of kernel operators where the eigenfunctions are approximated by the Nyström method. Below we briefly review SSGE. More details refer to Shi et al. (2018). Specifically, we denote the target gradient function to estimate by $\mathbf{g} : \mathcal{X} \to \mathbb{R}^d : \mathbf{g}(\mathbf{x}) = \nabla_{\mathbf{x}} \log q(\mathbf{x})$. The $i^{th}$ component of the gradient is $g_i(\mathbf{x}) = \nabla_{\mathbf{x}_i} \log q(\mathbf{x})$. We assume $g_1, \ldots, g_d \in L^2(\mathcal{X}, q)$. $\{\psi_j\}_{j \geq 1}$ denotes an orthonormal basis of $L^2(\mathcal{X}, q)$. We can expand $g_i(\mathbf{x})$ into the spectral series, i.e., $g_i(\mathbf{x}) = \sum_{j=1}^{\infty} \beta_{ij} \psi_j(\mathbf{x})$. The value of the $j^{th}$ eigenfunction $\psi_j$ at $\mathbf{x}$ can be approximated by the Nyström method (Xu et al., 2015). Due to the orthonormality of eigenfunctions $\{\psi_j\}_{j \geq 1}$, there is a constraint under the probability measure q(.): $\int \psi_i(\mathbf{x}) \psi_j(\mathbf{x}) q(\mathbf{x}) d\mathbf{x} = \delta_{ij}$, where $\delta_{ij} = \mathbb{1}[i = j]$. Based on this constraint, we can obtain the following equation for $\{\psi_j\}_{j \geq 1}$:

$$\int k(\mathbf{x}, \mathbf{y}) \psi(\mathbf{y}) q(\mathbf{y}) d\mathbf{y} = \mu \psi(\mathbf{x}), \tag{4}$$

where $k(.)$ is a kernel function. The left side of the above equation can be approximated by the Monte Carlo estimate using i.i.d. samples $\mathbf{x}^1, ..., \mathbf{x}^M$ from $q(.)$ : $\frac{1}{M} \mathbf{K} \psi \approx \mu \psi$, where $\mathbf{K}$ is the Gram Matrix and $\psi = \left[ \psi\left(\mathbf{x}^1\right), \ldots, \psi\left(\mathbf{x}^M\right) \right]^{\top}$. We can solve this eigenvalue problem by choose the $J$ largest eigenvalues $\lambda_1 \geq \cdots \geq \lambda_J$ for $\mathbf{K}$. $\boldsymbol{u}_j$ denotes the eigenvector of the Gram matrix. The approximation for $\{\psi_j\}_{j \geq 1}$ can be obtained combined with Eq. (4) as following: $\psi_j(\mathbf{x}) \approx \hat{\psi}_j(\mathbf{x}) = \frac{\sqrt{M}}{\lambda_j} \sum_{m=1}^{M} u_{jm} k\left(\mathbf{x}, \mathbf{x}^m\right)$.

Furthermore, based on the orthonormality of $\{\psi_j\}_{j \geq 1}$, we can easily obtain $\beta_{ij} = -\mathbb{E}_q \nabla_{\mathbf{x}_i} \psi_j(\mathbf{x})$. By taking derivative both sides of Eq. (4), we can show that:

$$\mu_j \nabla_{\mathbf{x}_i} \psi_j(\mathbf{x}) = \nabla_{\mathbf{x}_i} \int k(\mathbf{x}, \mathbf{y}) \psi_j(\mathbf{y}) q(\mathbf{y}) d\mathbf{y} = \int \nabla_{\mathbf{x}_i} k(\mathbf{x}, \mathbf{y}) \psi_j(\mathbf{y}) q(\mathbf{y}) d\mathbf{y}. \tag{5}$$

Then we can estimate as following:

$$\hat{\nabla}_{\mathbf{x}_i} \psi_j(\mathbf{x}) \approx \frac{1}{\mu_j M} \sum_{m=1}^{M} \nabla_{\mathbf{x}_i} k\left(\mathbf{x}, \mathbf{x}^m\right) \psi_j\left(\mathbf{x}^m\right). \tag{6}$$

Finally, by truncating the expansion to the first $J$ terms and plugging in the Nyström approximations of $\{\psi_j\}_{j \geq 1}$, we can get the score estimator:

$$\hat{g}_i(\mathbf{x}) = \sum_{j=1}^{J} \hat{\beta}_{ij} \hat{\psi}_j(\mathbf{x}), \qquad \hat{\beta}_{ij} = -\frac{1}{M} \sum_{m=1}^{M} \nabla_{\mathbf{x}_i} \hat{\psi}_j\left(\mathbf{x}^m\right). \tag{7}$$

In general, representation learning for large-scale datasets is usually costly in terms of storage and computation. For instance, the dimension of images in the STL-10 dataset is $96 \times 96 \times 3$ (i.e., the vector length is 27648). This makes it almost impossible to directly estimate the gradient of MI between the input and representation. To alleviate this problem, we introduce random projection (RP) (Bingham & Mannila, 2001) to reduce the dimension of $\mathbf{x}$.

We briefly review RP. More details refer to Bingham & Mannila (2001). RP projects the original $d$-dimensional data into a $k$-dimensional ($k << d$) subspace. Concretely, let matrix $X_{d \times N}$ denotes the original set of N $d$-dimensional data, the projection of the original data $X_{k \times N}^{RP}$ is obtained by introducing a random matrix $R_{k \times d}$ whose columns have unit length, as follows (Bingham & Mannila, 2001), $X_{k \times N}^{RP} = R_{k \times d} X_{d \times N}$. After RP, the Euclidean distance between two original data vectors can be approximated by the Euclidean distance of the projective vectors in reduced spaces:

$$\|\mathbf{x}_1 - \mathbf{x}_2\| \approx \sqrt{d/k} \|R\mathbf{x}_1 - R\mathbf{x}_2\|, \tag{8}$$

where $\mathbf{x}_1$ and $\mathbf{x}_2$ denote the two data vectors in the original large dimensional space.

Based on the principle of RP, we can derive a Salable Spectral Stein Gradient Estimator, which is an efficient high-dimensional score function estimator. One can show that the RBF kernel satisfies Steins identity (Liu & Wang, 2016). Shi et al. (2018) also shows that it is a promising choice for SSGE with a lower error bound. To reduce the computation of the kernel similarities of SSGE in high-dimensional settings, we replace the input of SSGE with a projections obtained by RP according to the approximation of Eq. (8) for the computation of the RBF kernel.

## 3 MUTUAL INFORMATION GRADIENT ESTIMATOR

As gradient estimation is a straightforward and effective method in optimization, we propose a gradient estimator for MI based on score estimation of implicit distributions, which is called Mutual Information Gradient estimator (MIGE). In this section, we focus on three most general cases of MI gradient estimation for representation learning, and derive the corresponding MI gradient estimator for these circumstances.

We outline the general setting of training an encoder to learn a representation. Let $\mathcal{X}$ and $\mathcal{Z}$ be the domain, and $E_\psi : \mathcal{X} \to \mathcal{Z}$ with parameters $\psi$ denotes a continuous and (almost everywhere) differentiable parametric function, which is usually a neural network, namely an encoder. $p(\mathbf{x})$ denotes the empirical distribution given the input data $\mathbf{x} \in \mathcal{X}$. We can obtain the representation of the input data through the encoder, $\mathbf{z} = E_\psi(\mathbf{x})$. $q_\psi(\mathbf{z})$ is defined as the marginal distribution induced by pushing samples from $p(\mathbf{x})$ through encoder $E_\psi(.)$ We also define $q_\psi(\mathbf{x}, \mathbf{z})$ as the joint distribution with $\mathbf{x}$ and $\mathbf{z}$, which is determined by encoder $E_\psi(.)$.

**Circumstance I**. Given that the encoder $E_\psi(.)$ is deterministic, our goal is to estimate the gradient of MI between input $\mathbf{x}$ and encoder output $\mathbf{z}$ w.r.t. the encoder parameters $\psi$. There is a close relationship between mutual information and entropy, which is as following:$I_\psi(\mathbf{x}; \mathbf{z}) = H(\mathbf{x}) + H_\psi(\mathbf{z}) - H_\psi(\mathbf{x}, \mathbf{z})$. Here $H(\mathbf{x})$ is data entropy and not relevant to $\psi$. The optimization of $I_\psi(\mathbf{x}, \mathbf{z})$ with parameters $\psi$ can neglect the entry $H(\mathbf{x})$. We decompose the gradient of the entropy of $q_\psi(\mathbf{z})$ and $q_\psi(\mathbf{x}, \mathbf{z})$ as (see Appendix A):

$$\nabla_\psi H(\mathbf{z}) = -\nabla_\psi \mathbb{E}_{q_\psi(\mathbf{z})}[\log q(\mathbf{z})], \quad \nabla_\psi H(\mathbf{x}, \mathbf{z}) = -\nabla_\psi \mathbb{E}_{q_\psi(\mathbf{x},\mathbf{z})}[\log q(\mathbf{x}, \mathbf{z})]. \tag{9}$$

Hence, we can represent the gradient of MI between input $\mathbf{x}$ and encoder output $\mathbf{z}$ w.r.t. encoder parameters $\psi$ as following:

$$\nabla_\psi I_\psi(\mathbf{x}; \mathbf{z}) = -\nabla_\psi \mathbb{E}_{q_\psi(\mathbf{z})}[\log q(\mathbf{z})] + \nabla_\psi \mathbb{E}_{q_\psi(\mathbf{x},\mathbf{z})}[\log q(\mathbf{x}, \mathbf{z})]. \tag{10}$$

However, this equation is intractable since an expectation w.r.t $q_\psi(\mathbf{z})$ is directly not differentiable w.r.t $\psi$. Roeder et al. (2017) proposed a general variant of the standard reparameterization trick for the variational evidence lower bound, which demonstrates lower-variance. To address above problem, we adapt this trick for MI gradient estimator in representation learning. Specifically, we can obtain the samples from the marginal distribution of $\mathbf{z}$ by pushing samples from the data empirical distribution $p(\mathbf{x})$ through $E_\psi(.)$ for representation learning. Hence we can reparameterize the representations variable $\mathbf{z} \sim q_\psi(\mathbf{z})$ using a differentiable transformation:$\mathbf{z} = E_\psi(\mathbf{x})$ with $\mathbf{x} \sim p(\mathbf{x})$, where the data empirical distribution $p(\mathbf{x})$ is independent of encoder parameters $\psi$. This reparameterization can rewrite an expectation w.r.t $q_\psi(\mathbf{z})$ and $q_\psi(\mathbf{x}, \mathbf{z})$ such that the Monte Carlo estimate of the expectation is differentiable w.r.t $\psi$.

Relying on this reparameterization trick, we can represent the gradient of MI w.r.t. encoder parameters $\psi$ in Eq. 10 as follows:

$$\begin{aligned} \nabla_\psi I_\psi(\mathbf{x}; \mathbf{z}) = &-\mathbb{E}_{q(\mathbf{x})}[\nabla_\mathbf{z} \log q(E_\psi(\mathbf{x}))\nabla_\psi E_\psi(\mathbf{x})] \\ &+ \mathbb{E}_{q(\mathbf{x})}[\nabla_{(\mathbf{x},\mathbf{z})} \log q(\mathbf{x}, E_\psi(\mathbf{x}))\nabla_\psi(\mathbf{x}, E_\psi(\mathbf{x}))], \end{aligned} \tag{11}$$

where the score function $\nabla_\mathbf{z} \log q_\psi(E_\psi(\mathbf{x}))$ can be estimated based on i.i.d. samples from an implicit density $q_\psi(\mathbf{E}_\psi(\mathbf{x}))$ (Shi et al., 2018; Song et al., 2019). The samples form the joint distribution $q_\psi(\mathbf{x}, \mathbf{z})$ are produced as following: we sample observations from empirical distribution $p(\mathbf{x})$; then the corresponding samples of $\mathbf{z}$ is obtained through $E_\psi(.)$. Hence we can also estimate $\nabla_{(\mathbf{x},\mathbf{z})} \log q(\mathbf{x}, E_\psi(\mathbf{x}))$ based on i.i.d. samples from $q_\psi(\mathbf{x}, E_\psi(\mathbf{x}))$. $\nabla_\psi E_\psi(\mathbf{x})$ and $\nabla_\psi(\mathbf{x}, E_\psi(\mathbf{x}))$ are directly computed with $\mathbf{x}$.

**Circumstance II.** Assume that we encode the input to latent data space $\mathbf{h} = C_\psi(\mathbf{x})$ that reflects useful structure in the data. Next, we summarize this latent variable mapping into final representations by the function $f_\psi$, $z = E_\psi(\mathbf{x}) = f_\psi \circ C_\psi(\mathbf{x})$. The gradient estimator of MI between $\mathbf{h}$ and $\mathbf{z}$ is represented by the data reparameterization trick as follows:

$$\nabla_\psi I_\psi(\mathbf{h}; \mathbf{z}) = \nabla_\psi H_\psi(\mathbf{h}) + \nabla_\psi H_\psi(\mathbf{z}) - \nabla_\psi H_\psi(\mathbf{h}, \mathbf{z}) \tag{12}$$

$$= -\mathbb{E}_{q(\mathbf{x})}[\nabla_\mathbf{z} \log q(E_\psi(\mathbf{x}))\nabla_\psi E_\psi(\mathbf{x})] - \mathbb{E}_{q(\mathbf{x})}[\nabla_\mathbf{h} \log q(C_\psi(\mathbf{x}))\nabla_\psi C_\psi(\mathbf{x})]$$

$$+ \mathbb{E}_{q(\mathbf{x})}[\nabla_{(\mathbf{h}, \mathbf{z})} \log q(C_\psi(\mathbf{x}), E_\psi(\mathbf{x}))\nabla_\psi (C_\psi\mathbf{x}, E_\psi(\mathbf{x}))]. \tag{13}$$

**Circumstance III.** Consider stochastic encoder function $E_\psi(., \boldsymbol{\epsilon})$ where $\boldsymbol{\epsilon}$ is an auxiliary variable with independent marginal $p(\boldsymbol{\epsilon})$. By utilizing data reparameterization trick. we can represent the gradient of the conditional entropy $H_\psi(\mathbf{z}|\mathbf{x})$ as follows (see Appendix A):

$$\nabla_\psi H_\psi(\mathbf{z}|\mathbf{x}) = -\mathbb{E}_{p(\mathbf{x})}[\mathbb{E}_{p(\boldsymbol{\epsilon})}[\nabla_{(\mathbf{z}|\mathbf{x})} \log q(E_\psi(\mathbf{x}, \boldsymbol{\epsilon})|\mathbf{x})\nabla_\psi E_\psi(\mathbf{x}, \boldsymbol{\epsilon})]], \tag{14}$$

where the term $\nabla_{(\mathbf{z}|\mathbf{x})} \log q(E_\psi(\mathbf{x}, \boldsymbol{\epsilon})|\mathbf{x})$ can be easily estimated by score estimation.

Based on the condition entropy gradient estimation in Eq. (14), the gradient estimator of MI between input and encoder output can be represented as following:

$$\nabla_\psi I_\psi(\mathbf{x}; \mathbf{z}) = \nabla_\psi H_\psi(\mathbf{z}) - \nabla_\psi H_\psi(\mathbf{z}|\mathbf{x}) \tag{15}$$

$$= -\mathbb{E}_{p(\mathbf{x})p(\boldsymbol{\epsilon})}[\nabla_\mathbf{z}[\log p(E_\psi(\mathbf{x}, \boldsymbol{\epsilon}))]\nabla_\psi E_\psi(\mathbf{x}, \boldsymbol{\epsilon})]$$

$$+ \mathbb{E}_{p(\mathbf{x})}[\mathbb{E}_{p(\boldsymbol{\epsilon})}[\nabla_{(\mathbf{z}|\mathbf{x})} \log q(E_\psi(\mathbf{x}, \boldsymbol{\epsilon})|\mathbf{x})\nabla_\psi E_\psi(\mathbf{x}, \boldsymbol{\epsilon})]]. \tag{16}$$

In practical MI optimization, we can construct MIGE of the full dataset based on mini-batch Monte Carlo estimates. We have provided an algorithm description for MIGE in Appendix B.

## 4 TOY EXPERIMENT

Recently, MINE and MINE-$f$ enable effective computation of MI in the continuous and high-dimensional settings. To compare with MINE and MINE-$f$, we evaluate MIGE in the correlated Gaussian problem taken from (Belghazi et al., 2018).

**Experimental Settings.** We consider two random variables $\mathbf{x}$ and $\mathbf{y}$ ($\mathbf{x}, \mathbf{y} \in \mathcal{R}^d$), coming from a $2d$-dimension multivariate Gaussian distribution. The component-wise correlation of $\mathbf{x}$ and $\mathbf{y}$ is defined as follows: $corr(\mathbf{x}_i, \mathbf{y}_i) = \delta_{ij}\rho$, $\rho \in (-1, 1)$, where $\delta_{ij}$ is Kronecker's delta and $\rho$ is the correlation coefficient. Since MI is invariant to smooth transformations of $\mathbf{x}, \mathbf{y}$, we only consider standardized Gaussian for marginal distribution $p(\mathbf{x})$ and $p(\mathbf{y})$. The gradient of MI w.r.t $\rho$ has the analytical solution: $\nabla_\rho I(\mathbf{x}; \mathbf{y}) = \frac{\rho d}{1-\rho^2}$. We apply MINE and MINE-$f$ to estimate MI of $\mathbf{x}, \mathbf{y}$ by sampling from the correlated Gaussian distribution and its marginal distributions, and the corresponding gradient of MI w.r.t $\rho$ can be computed by backpropagation implemented in Pytorch.

**Results.** Fig.1 presents our experimental results in different dimensions $d = \{5, 10, 20\}$. In the case of low-dimensional ($d = 5$), all the estimators give promising estimation of MI and its gradient. However, the MI estimation of MINE and MINE-$f$ are unstable due to its relying on a discriminator to produce estimation of the bound on MI. Hence, as showed in Fig.1, corresponding estimation of MI and its gradient is not smooth. As the dimension $d$ and the absolute value of correlation coefficient $|\rho|$ increase, MINE and MINE-$f$ are apparently hard to reach the True MI, and their gradient estimation of MI is thus high biased. This phenomenon would be more significant in the case of high-dimensional or large MI. Contrastively, MIGE demonstrates the significant improvement over MINE and MINE-$f$ when estimating MI gradient between twenty-dimensional random variables $\mathbf{x}, \mathbf{y}$. In this experiment, we compare our method with two baselines on an analyzable problem and find that the gradient curve estimated by our method is far superior to other methods in terms of smoothness and tightness in a high-dimensional and large-MI setting compared with MINE and MINE-$f$.

## 5 APPLICATIONS

To demonstrate the performance in downstream tasks, we deploy MIGE to Deep InfoMax (Hjelm et al., 2019) and Information Bottleneck (Tishby et al., 2000) respectively, namely replacing the original MI estimators with MIGE. We find that MIGE achieves higher and more stable classification accuracy, which indicating its good gradient estimation performance in practical applications.

Table 1: CIFAR-10 and CIFAR-100 classification accuracy (top 1) of downstream tasks compared with vanilla DIM. JSD and infoNCE are MI estimators, and PM denotes matching representations to a prior distribution (Hjelm et al., 2019).

| Model | CIFAR-10 | | | CIFAR-100 | | |
|---|---|---|---|---|---|---|
| | conv | fc(1024) | Y(64) | conv | fc(1024) | Y(64) |
| DIM (JSD) | 55.81% | 45.73% | 40.67% | 28.41% | 22.16% | 16.50% |
| DIM (JSD + PM) | 52.2% | 52.84% | 43.17% | 24.40% | 18.22% | 15.22% |
| DIM (infoNCE) | 51.82% | 42.81% | 37.79% | 24.60% | 16.54% | 12.96% |
| DIM (infoNCE + PM) | 56.77% | 49.42% | 42.68% | 25.51% | 20.15% | 15.35% |
| MIGE | **57.95%** | **57.09%** | **53.75%** | **29.86%** | **27.91%** | **25.84%** |

Table 2: STL-10 classification accuracy (top 1) of downstream tasks compared with vanilla DIM. The dimension of STL-10 images (27648) results in exorbitant computational cost. Random Projection (RP) is applied to reduce the dimension.

| Model | STL-10 | | |
|---|---|---|---|
| | conv | fc(1024) | Y(64) |
| DIM (JSD) | 42.03% | 30.28% | 28.09% |
| DIM (infoNCE) | 43.13% | 35.80% | 34.44% |
| MIGE | unaffordable computational cost | | |
| MIGE + RP to 512d | 52.00% | 48.14% | 44.89% |

Figure 2: STL-10 Y(64) classification accuracy (top 1) with different RP dimension.

## 5.1 DEEP INFOMAX

Discovering useful representations from unlabeled data is one core problem for deep learning. Recently, a growing set of methods is explored to train deep neural network encoders by maximizing the mutual information between its input and output. A number of methods based on tractable variational lower bounds, such as JSD and infoNCE, have been proposed to improve the estimation of MI between high dimensional input/output pairs of deep neural networks (Hjelm et al., 2019). To compare with JSD and infoNCE, we expand the application of MIGE in unsupervised learning of deep representations based on the InfoMax principle.

**Experimental Settings.** For consistent comparison, we follow the experiments of Deep Info-Max(DIM)[1] to set the experimental setup as in Hjelm et al. (2019). We test DIM on image datasets CIFAR-10, CIFAR-100 and STL-10 to evaluate our MIGE. For the high-dimensional images in STL-10, directly applying SSGE is almost impossible since it results in exorbitant computational cost. Our proposed Scalable SSGE is applied, to reduce the dimension of images and achieve reasonable computational cost. As mentioned in Hjelm et al. (2019), non-linear classifier is chosen to evaluate our representation, After learning representation, we freeze the parameters of the encoder and train a non-linear classifier using the representation as the input. The same classifiers are used for all methods. Our baseline results are directly copied from Hjelm et al. (2019) or by running the code of author.

**Results.** As shown in Table 1, MIGE outperforms all the competitive models in DIM experiments on CIFAR-10 and CIFAR-100. Besides the numerical improvements, it is notable that our model have the less accuracy decrease across layers than that of DIM(JSD) and DIM(infoNCE). The results indicate that, compared to variational lower bound methods, MIGE gives more favorable gradient direction, and demonstrates more power in controlling information flows without significant loss. With the aid of Random Projection, we could evaluate on bigger datasets, e.g., STL-10. Table 2 shows the result of DIM experiments on STL-10. We can observe significant improvement over the baselines when RP to 512d. Note that our proposed gradient estimator can also be extended to the multi-view setting(i.e., with local and global features) of DIM, it is beyond the scope of this paper. More discussions refer to Appendix C.

---

[1]Codes available at `https://github.com/rdevon/DIM`

**Ablation Study.** To verify the effect of different dimensions of Random Projection on classification accuracy in DIM experiments, we conduct an ablation study on STL-10 with the above experimental settings. Varying RP dimension $k \in \{16, 32, 64, 128, 256, 512, 1024\}$, we measure the classification accuracy of Y(64) which is shown in Fig.2. We find that the classification accuracy increases with RP dimension from 16 to 128. After that, the approximation in Equ.(8) with the further increase of the RP dimension reaches saturation, while bringing extra computational costs.

## 5.2 INFORMATION BOTTLENECK

Information Bottleneck (IB) has been widely applied to a variety of application domains, such as classification (Tishby & Zaslavsky, 2015; Alemi et al., 2017; Chalk et al., 2016; Kolchinsky et al., 2017), clustering (Slonim & Tishby, 2000), and coding theory and quantization (Zeitler et al., 2008; Courtade & Wesel, 2011). In particular, given the input variable $\mathbf{x}$ and the target variable $\mathbf{y}$, the goal of the IB is to learn a representation of $\mathbf{x}$ (denoted by the variable $\mathbf{z}$) that satisfies the following characteristics:

1) $\mathbf{z}$ is sufficient for the target $\mathbf{y}$, that is, all information about target $\mathbf{y}$ contained in $\mathbf{x}$ should also be contained in $\mathbf{z}$. In optimization, it should be

2) $\mathbf{z}$ is minimal. In order not to contain irrelevant information that is not related to $\mathbf{y}$, $\mathbf{z}$ is required to contain the smallest information among all sufficient representations.

The objective function of IB is written as follows:

$$\max I(\mathbf{z}; \mathbf{y}), \quad \text{s.t. } I(\mathbf{z}; \mathbf{x}) \leq c. \tag{17}$$

Equivalently, by introducing a Lagrangian multiplier $\beta$, the IB method can maximize the following objective function: $G_{IB} = I(\mathbf{z}; \mathbf{y}) - \beta I(\mathbf{z}; \mathbf{x})$. Further, it is generally acknowledged that $I(\mathbf{z}; \mathbf{y}) = H(\mathbf{y}) - H(\mathbf{y}|\mathbf{z})$, and $H(\mathbf{y})$ is constant. Hence we can also minimize the objective function of the following form:

$$L_{IB} = H(\mathbf{y}|\mathbf{z}) + \beta I(\mathbf{z}; \mathbf{x}), \tag{18}$$

where $\beta \geq 0$ plays a role in trading off the sufficiency and minimality. Note that the above formulas omit the parameters for simplicity.

To overcome the intractability of MI in the continuous and high-dimension setting, Alemi et al. (2017) presents a variational approximation to IB, which adopts deep neural network encoder to produce a conditional multivariate normal distribution, called Deep Variational Bottleneck (DVB). Rencently, DVB is exploited to restrict the capacity of discriminators in GANs (Peng et al., 2019). However, a tractable density is required for the approximate posterior in DVB due to their reliance on a variational approximation while MIGE does not.

To evaluate our method, we compare MIGE-IB with DVB and MINE-IB in IB application. We demonstrate an implementation of the IB objective on permutation invariant MNIST using MIGE.

**Experiments.** For consistent comparison, we adopt the same architecture and empirical settings used in Alemi et al. (2017) except that the initial learning rate of 2e-4 is set for Adam optimizer, and exponential decay with decaying rate by a factor of 0.96 was set for every 2 epochs. The implementation of DVB is available from its authors[2]. Under these experimental settings, we use our MI Gradient Estimator to replace the MI estimator in DVB experiment. The threshold of score function's Stein gradient estimator is set as 0.94. The threshold is the hyper-parameter of Spectral Stein Gradient Estimator (SSGE), and it is used to set the kernel bandwidth of RBF kernel. Our results can be seen in Table 3 and it manifests that our proposed MIGE-IB outperforms DVB and MINE-IB.

---

[2]https://github.com/alexalemi/vib_demo

Table 3: Permutation-invariant MNIST misclassification rate. Datas except our model are cited from Belghazi et al. (2018)

| Model | Misclass rate |
|---|---|
| Baseline | 1.38% |
| Dropout | 1.34% |
| Confidence penalty | 1.36% |
| Label Smoothing | 1.4% |
| DVB | 1.13% |
| MINE-IB | 1.11% |
| MIGE-IB (ours) | **1.05%** |

## 6 CONCLUSION

In this paper, we present a gradient estimator, called Mutual Information Gradient Estimator (MIGE), to avoid the various problems met in direct mutual information estimation. We manifest the effectiveness of gradient estimation of MI over direct MI estimation by applying it in unsupervised or supervised representation learning. Experimental results have indicated the remarkable improvement over MI estimation in the Deep InfoMax method and the Information Bottleneck method.

## ACCKNOWLEDGEMENT

This work was partially funded by the National Key R&D Program of China (No. 2018YFB1005100 & No. 2018YFB1005104).

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

## A    DERIVATION OF GRADIENT ESTIMATES FOR ENTROPY

**Unconditional Entropy** Given that the encoder $E_\psi(.)$ is deterministic, our goal is to optimize the entropy $H(q) = -\mathbb{E}_q \log q$, where $q$ is short for the distribution $q_\psi(\mathbf{z})$ of the representation $\mathbf{z}$ w.r.t. its parameters $\psi$. We can decompose the gradient of the entropy of $q_\psi(\mathbf{z})$ as:

$$\nabla_\psi H(z) = -\nabla_\psi \mathbb{E}_{q_\psi(\mathbf{z})}[\log q(\mathbf{z})] - \mathbb{E}_{q(\mathbf{z})}[\nabla_\psi \log q_\psi(\mathbf{z})], \tag{19}$$

The second term on the right side of the equation can be calculated:

$$\mathbb{E}_{q(\mathbf{z})}[\nabla_\psi \log q_\psi(\mathbf{z})] = \mathbb{E}_{q(\mathbf{z})}[\nabla_\psi q_\psi(\mathbf{z}) \times \frac{1}{q(\mathbf{z})}] = \int \nabla_\psi q_\psi(\mathbf{z}) d\mathbf{z} = \nabla_\psi \int q_\psi(\mathbf{z}) d\mathbf{z} = 0. \tag{20}$$

Therefore, the gradient of the entropy of $q_\psi(\mathbf{z})$ becomes

$$\nabla_\psi H(z) = -\nabla_\psi \mathbb{E}_{q_\psi(\mathbf{z})}[\log q(\mathbf{z})]. \tag{21}$$

**Conditional Entropy** Consider nondeterministic encoder function $E_\psi(., \epsilon)$ where $\epsilon$ is an auxiliary variable with independent marginal $p(\epsilon)$. The distribution $q_\psi(z|x)$ is determined by $\epsilon$ and the encoder parameters $\psi$. The auxiliary variable $\epsilon$ introduces randomness to the encoder. First, we decompose the gradients of Conditional Entropy as following:

$$\begin{aligned}
\nabla_\psi H(\mathbf{z}|\mathbf{x}) &= -\nabla_\psi \int p_\psi(\mathbf{z}, \mathbf{x}) \log p_\psi(\mathbf{z}|\mathbf{x}) dz dx \\
&= -\mathbb{E}_{p(\mathbf{x})}[\nabla_\psi \int p_\psi(\mathbf{z}|\mathbf{x}) \log p_\psi(\mathbf{z}|\mathbf{x}) dz] \\
&= -\mathbb{E}_{p(\mathbf{x})}[\nabla_\psi \mathbb{E}_{p_\psi(\mathbf{z}|\mathbf{x})}[\log p(\mathbf{z}|\mathbf{x})] + \int p(\mathbf{z}|\mathbf{x}) \nabla_\psi \log p_\psi(\mathbf{z}|\mathbf{x}) dh] \\
&= -\mathbb{E}_{p(\mathbf{x})}[\nabla_\psi \mathbb{E}_{p_\psi(\mathbf{z}|\mathbf{x})}[\log p(\mathbf{z}|\mathbf{x})] + \int \nabla_\psi p_\psi(\mathbf{z}|\mathbf{x}) dh] \\
&= -\mathbb{E}_{p(\mathbf{x})}[\nabla_\psi \mathbb{E}_{p_\psi(\mathbf{z}|\mathbf{x})}[\log p(\mathbf{z}|\mathbf{x})] - \nabla_\psi \int p_\psi(\mathbf{h}, \mathbf{x}) dh dx] \\
&= -\mathbb{E}_{p(\mathbf{x})}[\nabla_\psi \mathbb{E}_{p_\psi(\mathbf{z}|\mathbf{x})}[\log p(\mathbf{z}|\mathbf{x})]].
\end{aligned} \tag{22}$$

Note that $\mathbf{z} = E_\psi(\mathbf{x}, \epsilon)$, such that we can apply reparameterization trick to the gradient estimator of conditional entropy in Eq. (22),

$$H_\psi(\mathbf{z}|\mathbf{x}) = -\mathbb{E}_{p(\mathbf{x})}[\mathbb{E}_{p(\epsilon)}[\nabla_{(\mathbf{z}|\mathbf{x})} \log q(E_\psi(\mathbf{x}, \epsilon)|\mathbf{x}) \nabla_\psi E_\psi(\mathbf{x}, \epsilon)]]. \tag{23}$$

## B    MIGE ALGORITHM DESCRIPTION

The algorithm description of our proposed MIGE is stated in Algorithm 1.

---
**Algorithm 1** MIGE (Circumstance I)
---
**1. Sampling**:
  Draw $n$ samples from the data distribution $p(x)$, $n$ denotes mini-batch size,
  then compute the corresponding output of the encoder
  $(\mathbf{x}^{(1)}, \mathbf{z}^{(1)}), \cdots, (\mathbf{x}^{(n)}, \mathbf{z}^{(n)}) \sim q_\psi(\mathbf{x}, \mathbf{z})$
  $\mathbf{z}^{(1)}, \cdots, \mathbf{z}^{(n)} \sim q_\psi(\mathbf{z})$
**2. Estimate the score function**:
  $\nabla_\mathbf{z} \log q_\psi(\mathbf{z}^{(i)}) \leftarrow \text{SSGE}(\mathbf{z}^{(1)}, \cdots, \mathbf{z}^{(n)})$
  $\nabla_{(\mathbf{x},\mathbf{z})} \log q_\psi(\mathbf{x}^{(i)}, \mathbf{z}^{(i)}) \leftarrow \text{SSGE}((\mathbf{x}^{(1)}, \mathbf{z}^{(1)}), \cdots, (\mathbf{x}^{(n)}, \mathbf{z}^{(n)}))$
**3. Estimate the entropy gradient**:
  $\nabla_\psi H(\mathbf{z}) \leftarrow -\frac{1}{n} \sum_{i=1}^{n} [\nabla_\psi \mathbf{z}^{(i)} \nabla_\mathbf{z} \log q_\psi(\mathbf{z}^{(i)})]$
  $\nabla_\psi H(\mathbf{x}, \mathbf{z}) \leftarrow -\frac{1}{n} \sum_{i=1}^{n} [\nabla_\psi(\mathbf{x}^{(i)}, \mathbf{z}^{(i)}) \nabla_{(\mathbf{x},\mathbf{z})} \log q_\psi(\mathbf{x}^{(i)}, \mathbf{z}^{(i)})]$
**4. Estimate the MI gradient**:
  $\nabla_\psi I(\mathbf{x}; \mathbf{z}) \leftarrow \nabla_\psi H(\mathbf{z}) - \nabla_\psi H(\mathbf{x}; \mathbf{z})$
---

## C  DISCUSSION ON DIM(L)

DIM(L) (Hjelm et al., 2019) is the state-of-the-art unsupervised model for representaion learning, which maximizes the average MI between the high-level representation and local patches of the image, and achieve an even higher classification accuracy than supervised learning. As shown in Table 4, we apply MIGE into DIM(L) and surprisingly find there is a significant performance gap to DIM(L).

To our knowledge, the principle of DIM(L) is still unclear. Tschannen et al. (2019) argues that maximizing tighter bounds in DIM(L) can lead to worse results, and the success of these methods cannot be attributed to the properties of MI alone, and they strongly depend on the inductive bias in both the choice of feature extractor architectures and the parameterization of the employed MI estimators. For MIGE, we are investigating the behind reasons, e.g., to investigate the distributions of the patches.

Table 4: CIFAR-10 and CIFAR-100 classification accuracy (top 1) of downstream tasks compared with vanilla DIM(L).

| Model | CIFAR-10 | | | CIFAR-100 | | |
|---|---|---|---|---|---|---|
| | conv | fc(1024) | Y(64) | conv | fc(1024) | Y(64) |
| DIM(L) (JSD) | 72.16% | 67.99% | 66.35% | 41.65% | 39.60% | 39.66% |
| DIM(L) (JSD + PM) | 73.25% | 73.62% | 66.96% | 48.13% | 45.92% | 39.6% |
| DIM(L) (infoNCE) | 75.05% | 70.68% | **69.24%** | 44.11% | 42.97% | **42.74%** |
| DIM(L) (infoNCE + PM) | **75.21%** | **75.57%** | 69.13% | **49.74%** | **47.72%** | 41.61% |
| MIGE | 59.72% | 56.14% | 54.01% | 30.0% | 28.96% | 27.65% |

