# OpenReview forum: "Mutual Information Gradient Estimation for  Representation Learning"
_ICLR.cc/2020/Conference — Accept (Poster)_

### Official Review · AnonReviewer1 · 2019-10-22
**Official Blind Review #1**

**Rating:** 8

**Review:**

This paper proposes MIGE---a novel estimator of the mutual information (MI) gradient, based on estimating the score function of an implicit distribution. To this end, the authors employ the spectral Stein gradient estimator (SSGE) and propose its scalable version based on random projections of the original input. The theoretical advantages of the method are presented using a toy experiment with correlated Gaussian random variables, where both the mutual information and its gradient can be computed analytically. In this setting, MIGE provides gradient estimates that are less biased and smoother than baselines. The method is also evaluated on two more complicated tasks: unsupervised representation learning on Cifar-10 and CIfar-100 via DeepInfoMax (DIM) and classification on MNIST with Information Bottleneck (IB), where MIGE outperforms all baselines by a significant margin.

I recommend ACCEPTing this paper. This work discusses a vital problem and proposes a novel, well-motivated, principled and very performant solution; additionally, it demonstrates the broad applicability of the introduced method. While the proposed technique consists of previously known building blocks (spectral Stein gradient estimator and random projections), it is cleverly applied in a novel context of estimating MI gradients.

While the paper is solid, I believe that it could be improved in the following ways. Firstly, I would like to see 1) more extensive and 2) larger-scale evaluations. In the DIM experiment, 1) would correspond to trying the DIM(L) approach, which maximises patch-wise MI. In fact, I strongly recommend including this experiment as it corresponds to and could improve current state-of-the-art. If it turns out that MIGE does not work well on DIM(L), then this would correspond to a serious issue with the method. In this experiment, 1) would also include providing other metrics for learned representations. It would be much more convincing to include estimates of true mutual information (e.g. InfoNCE bound evaluated with a large number of samples [1]) and showing that MIGE can attain higher values than baselines. 2) would correspond to evaluation on bigger datasets: (tiny) ImageNet and STL-10 dataset. Also, the toy experiment would benefit from a higher-dimensional setting (e.g. d=256 to d=1024), since these are often used in practice.
Secondly, the paper is sloppily-written, which quite a few grammar and stylistic mistakes (e.g. sentence in sec 3, paragraph 2: “we assume obtain to…”, which starts with a lower-case letter and doesn’t make sense). Finally, the paper would benefit from the following clarifications: 1) explain what is the Nystr\:om method, 2) provide either a proof or a citation for eq (19); also the error bound for SSGE should be provided for the paper to be self-contained, 3) explain the difference between q_\psi and p_\psi, which seem to be used interchangeably.

Additional remarks:
Sec 2.2, 2), “streamlining” is unclear
Circumstances 2 and 3 can be quite easily derived from circumstance 1; also they are not evaluated empirically; it would be nice to have experiments for them, and they can be moved to the appendix in case of lack of space
Eq (19) while nice, seem to bear no significance for the proposed method and the rest of the paper; consider removing it
Section 4.2 paragraph 2: “shrinking” for different layers wasn’t mentioned before, and is not immediately clear what it means; the reader needs to be intimately familiar with the DIM paper to understand.
Section 4.3 mentions “threshold” for stein gradient estimator, which was not mentioned before. Please explain what it is.
Equations (8-10) are just simple derivations and are not necessary; it would be enough to provide Eq (10).
The authors talk about MINE, which optimizes the InfoNCE bound [1], which is also used in DIM and CPC [2]. I strongly encourage the authors to cite [1] and [2] and mention them in the related works. Additionally, it would be clear if Figure 1 and related references and description used “InfoNCE” instead of “MINE” as the name of the method since InfoNCE is an estimator and MINE is just a particular implementation of the method.

[1] Poole et. al., “On variational bounds of mutual information”, ICML 2019.
[2] van den Oord et. al, “Representation Learning with Contrastive Predictive Coding“, arXiv 2018.

**Experience Assessment:**

I have read many papers in this area.

**Review Assessment: Checking Correctness Of Derivations And Theory:**

I carefully checked the derivations and theory.

**Review Assessment: Checking Correctness Of Experiments:**

I carefully checked the experiments.

**Review Assessment: Thoroughness In Paper Reading:**

I read the paper thoroughly.

---

> ### Author Response · Authors · 2019-11-13
> **Response to Reviewer 1**
>
> We thank the reviewer for the acknowledgment of our paper and the valuable suggestions. We are glad that the reviewer found the work to be novel and well-motivated. We will carefully revise our manuscript and hire professional copy editors to proofread our paper.
>
> 1) R: Thanks for pointing out the recommendation to apply MIGE on DIM(L) as this is a good suggestion. Due to limited responce time and computational resources, we are still in the process of experimenting and analyzing preliminary results.  In addition, we note that the literature has argued that maximizing tighter bounds in DIM(L) leads to worse results [2]. We will consider this as future work and discuss these new findings in a new paper.
>
> We cannot provide the value of MI to evaluate the representation, because MIGE directly estimates the gradient of MI to optimize MI, rather than estimating the value of MI.
>
> 2) R: For comments on evaluating on larger datasets, we have evaluated our method on a large scale dataset, i.e., STL-10. The results can be found in the following table.
>
> Table: Classification accuracy (top 1) results on STL-10.
> RP denotes Random Projection.
> .----------------------------------------------------------------
> .							STL-10
> .					conv	fc		Y
> .----------------------------------------------------------------
> .DIM(JSD)			42.03	30.28	28.09
> .DIM(infoNCE)		43.13	35.80	34.44
> .----------------------------------------------------------------
> .MIGE			unaffordable computational cost
> .MIGE+RP to 1024d	49.08	40.09	38.95
> .MIGE+RP to 512d	49.89	41.05	38.56
> .MIGE+RP to 256d	49.91	40.24	38.83
> .----------------------------------------------------------------
>
> R: The setting of our toy experiment follows the literature [3, 4], where the  highest-dimensions  is set to 20. We will add the evaluation on the toy experiment in  high dimensional settings (e.g., 256, 1024).
>
>
> Q1：Circumstances 2 and 3 can be quite easily derived from circumstance 1; also they are not evaluated empirically; it would be nice to have experiments for them, and they can be moved to the appendix in case of lack of space
>
> R: MIGE in Circumstances 3 is applied to the Information Bottleneck experiments in Section 4.3. We will add explanations in Section 4.3.
>
> Q2: Section 4.2 paragraph 2: “shrinking” for different layers wasn’t mentioned before, and is not immediately clear what it means; the reader needs to be intimately familiar with the DIM paper to understand.
>
> R: We will add the detail description to make the reader easy to understand "shrinking".
>
> Q3: explain the difference between q_\psi and p_\psi, which seem to be used interchangeably.
>
> R: p_\psi is a typo and should be instead of q_\psi. p(.) means that this distribution is uncorrelated to the encoder parameters \psi. q_\psi(.) is determined by the encoder parameters \psi(.)
>
> Q4: Section 4.3 mentions “threshold” for stein gradient estimator, which was not mentioned before. Please explain what it is.
>
> R:“threshold” is a hyperparameter of Spectral Stein Gradient Estimator (SSGE), and it is used to set the kernel bandwidth of RBF kernel.
>
> Q5: The authors talk about MINE, which optimizes the InfoNCE bound [1], which is also used in DIM and CPC [2]. I strongly encourage the authors to cite [1] and [2] and mention them in the related works. Additionally, it would be clear if Figure 1 and related references and description used “InfoNCE” instead of “MINE” as the name of the method since InfoNCE is
>
> R: Indeed, we have cited these two papers. MINE optimizes the Donsker-Varadhan representation rather than the InfoNCE bound. MINE-f optimizes the f-divergence representation. InfoNCE is a different estimator with high bias. For more details please refer to [3, 4].
>
>
>
> [1] R Devon Hjelm, Alex Fedorov, Samuel Lavoie-Marchildon, Karan Grewal, Phil Bachman, Adam Trischler, and Yoshua Bengio. Learning deep representations by mutual information estimation and maximization. In International Conference on Learning Representations, 2019.
> [2] On mutual inforamtion maximization for representations, https://openreview.net/forum?id=rkxoh24FPH.
> [3] Ishmael Belghazi, Aristide Baratin, Sai Rajeswar, Sherjil Ozair, Yoshua Bengio, Aaron Courville, and R Devon Hjelm. Mine: mutual information neural estimation. arXiv preprint arXiv:1801.04062, ICML, 2018.
> [4] Ben Poole, Sherjil Ozair, Aaron van den Oord, Alex Alemi, and George Tucker. On variational bounds of mutual information. In International Conference on Machine Learning, 2019.

---

> > ### Comment · AnonReviewer1 · 2019-11-13
> > **response**
> >
> > > In addition, we note that the literature has argued that maximizing tighter bounds in DIM(L) leads to worse results [2].
> >
> > The above might be true for a specific setting when a more expressive critic is used to estimate the InfoNCE bound but has no significance to the paper. As you point out yourself, you are not estimating MI, but only its gradient. If using MIGE for DIM(L) leads to worse results than in the original paper, then it should be clearly stated and discussed -- it will not diminish the value of this paper.
> >
> > > We cannot provide the value of MI to evaluate the representation, because MIGE directly estimates the gradient of MI to optimize MI, rather than estimating the value of MI.
> >
> > What you can do is to train a separate critic to estimate InfoNCE as in [2], and evaluate InfoNCE with a large batch size. This result would be very helpful as it would shed light on how much information is actually stored in representations learned with MIGE.
> >
> > Thanks for the experiments on STL-10 and further clarification. I'm hoping to read a response to the above comments before the end of the discussion period.

---

> > > ### Author Response · Authors · 2019-11-14
> > > **Response to Reviewer 1**
> > >
> > >
> > >
> > > R1: We have conducted experiments of applying MIGE for DIM(L) in the CIFAR dataset and we show the results in the following Table. Surprisingly, there is a significant gap to DIM(L).  To analyze this result, we find during training of CIFAR-10, the testing accuracy gets stable after reaching over 50%, while the training accuracy soon reaches 99%.  This may suggest that some regularization technique may be needed to the gradient estimation of DIM(L).
> > >
> > > To our knowledge, the principle of DIM(L) is still  unclear.  As argued in [2],   the success of these methods cannot be attributed to the properties of MI alone, and they strongly depend on the inductive bias in both the choice of feature extractor architectures and the parameterization of the employed MI estimators.
> > >
> > > For MIGE, we are investigating the behind reason, e.g., to investigate the distribution of the patches.
> > >
> > > .---------------------------------------------------------------------------------------------------
> > > .							CIFAR10					CIFAR100
> > > .					conv	fc		Y		conv	fc		Y
> > > .---------------------------------------------------------------------------------------------------
> > > .DIM(L)(JSD)			72.16	67.99	66.35	41.65	39.60	39.66
> > > .DIM(L)(JSD+PM)		73.25	73.62	66.96	48.13	45.92	39.60
> > > .DIM(L)(infoNCE)		75.05	70.68	69.24	44.11	42.97	42.74
> > > .DIM(L)(infoNCE+PM)	75.21	75.57	69.13	49.74	47.72	41.61
> > > .---------------------------------------------------------------------------------------------------
> > > .MIGE(L)			59.72	56.14	54.01	30.00	28.96	27.65
> > > .---------------------------------------------------------------------------------------------------
> > >
> > >
> > >
> > >
> > > R2: Thanks for providing the insight. We can use MINE (as used in [1], and MINE is better than InfoNCE due to the high bias of InforNCE) for estimating MI stored in representations. Due to the limited response time, we will add the metric in the camera-ready.

---

> > > > ### Comment · AnonReviewer1 · 2019-11-14
> > > > **Thanks for the response**
> > > >
> > > > Thanks for the response. It would be great if you could add the DIM(L) results to the paper and discuss the issues, e.g. in the appendix.
> > > >
> > > > Your paper is a piece of some really solid work. I strongly recommend acceptance!

---

> > > > > ### Author Response · Authors · 2019-11-15
> > > > > **Response to Reviewer 1**
> > > > >
> > > > > Thank you for recommence.
> > > > > We will add the statement related to [1] to the appendix in the revision.

---

> > ### Comment · AnonReviewer1 · 2019-11-13
> > **proofreading**
> >
> > >  We will carefully revise our manuscript and hire professional copy editors to proofread our paper.
> >
> > Thanks; I don't think professional service is necessary. Please just use spell-check and have someone proofread the paper.

---

### Official Review · AnonReviewer3 · 2019-10-31
**Official Blind Review #3**

**Rating:** 6

**Review:**

This paper works out estimators for the gradient of Mutual Information (MI). The focus is on its recent popular use for representation learning. The insight the authors provide is to see encoding the representation as a ‘reparametrization’ of the data. This insight enables mathematical tools from the literature on ‘pathwise derivatives’. With gradients on the MI, one can estimate models that aim to maximize this quantity. For example in unsupervised learning one can learn representations for downstream tasks. This is shown in Table 1. Another application in supervised learning is the Information Bottleneck. This shown in Table 2.

Three points for review:
1)
The estimator for the gradient is shown at first on a toy task. Take a correlated Gaussian distribution and estimate gradients of the MI. The correlated Gaussian has an analytical form of MI, which makes this a useful experiment. The paper claims that this estimator ‘provides a tighter and smoother’ gradient estimate. I don’t see how this experiment and this claim tie together. Could the tightness or smoothness be quantified? It seems the MIGE has a lower variance, could empirical results or bounds on the variance be obtained?

Moreover, this plot concerns random quantities, whereas we see only one realization. Both the MIGE and the MINE hold under expectation of samples from the data distribution. This is a toy example where we can sample infinitely from the data distribution. That means we can either a) plot more samples or b) obtain (empirical) error bounds on the gradient under these sampling distributions.

2)
The major experimental result in the paper shows advantage of the gradient estimator in Transfer learning. Specifically, the authors compare against the recent DIM of Hjelm et al 2019. The authors train a quote ‘small fully connected neural network classifier’. However, the work of Hjelm trains a linear SVM on the representations. It is not clear where the increase in performance originates. Is it the improved representations (as obtained by using MIGE) or is it the change classifier?

3)
One contribution of the paper is to make the gradient estimator work in high dimensions. To this end, the authors propose Random Projections. It is not clear how this approximation influences the results. An experiment regarding this topic would make the point clearer. Is RP used in the current experiments? Then how does this influence the results? Is the RP used for computational purposes? Then can we quantify the gain in computation?

Minor comments:
  *’In practice, we do not care about MI estimation’. Please explain further or refer to previous work.
  *’In optimization, it should be achieved by maximizing the information between z and z.’ (Section 2). Two points
    1. The information ‘between z and z’ is probably a typo?
    2. How does sufficiency relate to an optimization problem? Doesn’t sufficiency mean in this context I(X;Y)=I(Z;Y)?
  * Equation 12: In the part $\nabla_psi (x, E_\psi(x))$. Why do we take gradient w.r.t. x? It seems to me that the reparametrization is a function of x only via $E_\psi(x)$. If not, then please explain what this tuple means.
  * Table 1 has no units. How to interpret the numbers in this table?
  * Section 4.3, authors note their experiment is ‘a little bit different’ from other related research. How and what exactly is different?

Typographic comments
  *Just below eqn17, ‘minibatche’ => ‘mini-batch’ or ‘mini batch’
  *Section 4.2 ‘images classification’ => ‘image classification’
  *’However A tractable density is’ => ‘However, a tractable density is’
  *’Estimating gradients of MI than’ -> ‘Estimating gradients of MI rather than’
  *Section 3, circumstance 1 ‘representation’ => ‘represent’


**Experience Assessment:**

I have read many papers in this area.

**Review Assessment: Checking Correctness Of Derivations And Theory:**

I assessed the sensibility of the derivations and theory.

**Review Assessment: Checking Correctness Of Experiments:**

I assessed the sensibility of the experiments.

**Review Assessment: Thoroughness In Paper Reading:**

I read the paper thoroughly.

---

> ### Author Response · Authors · 2019-11-13
> **Response to Reviewer 3 (1)**
>
> Thank you for acknowledging that our idea is interesting and the results are encouraging. We will revise our paper according to the comments and hire a copy-editor to carefully polish the writing.
>
>
> Q1:The estimator for the gradient is shown at first on a toy task. Take a correlated Gaussian distribution and estimate gradients of the MI. The correlated Gaussian has an analytical form of MI, which makes this a useful experiment. The paper claims that this estimator ‘provides a tighter and smoother’ gradient estimate. I don’t see how this experiment and this claim tie together.
>
> R: In terms of " tighter and smoother’ gradient estimate", Fig.1 shows that the proposed MI Gradient Estimator is the tighter and smoother in a high-dimensional and large MI setting than the competitors. MINE and MINE-f have high variance, and suffer from unstable estimation due to their dependence on the discriminator used to estimate the bound of mutual information. The toy experiment shows that the MIGE has a lower variance.
>
> Q2: Could the tightness or smoothness be quantified? It seems the MIGE has a lower variance, could empirical results or bounds on the variance be obtained?
> Moreover, this plot concerns random quantities, whereas we see only one realization. Both the MIGE and the MINE hold under expectation of samples from the data distribution. This is a toy example where we can sample infinitely from the data distribution. That means we can either a) plot more samples or b) obtain (empirical) error bounds on the gradient under these sampling distributions.
>
>
> R:The correlated Gaussian experiment is taken from [1] which did not provide quantitive measures. However, we agree with the reviewer that it is a very nice idea to quantify the tightness and smoothness. We will try to add the qualification of tightness and smoothness to make our conclusion more convincing. Thank you for your valuable suggestion again.
>
> Q3: The authors train a quote ‘small fully connected neural network classifier’. However, the work of Hjelm trains a linear SVM on the representations.Is it the improved representations (as obtained by using MIGE) or is it the change classifier?
>
> R:  For consistent comparison, the baseline of DIM and our proposed MIGE are all based on non-linear classification, which is also mentioned in [2]. We did not include any result of linear SVM for DIM or any other methods.
> The same classifiers are used for all methods. Our baseline results are directly copied from [2] or by running the code in https://github.com/rdevon/DIM. We haven’t changed the code of the non-linear classification.

---

> > ### Author Response · Authors · 2019-11-13
> > **Response to Reviewer 3  (2)**
> >
> >
> >
> > Q4: One contribution of the paper is to make the gradient estimator work in high dimensions. To this end, the authors propose Random Projections. It is not clear how this approximation influences the results. An experiment regarding this topic would make the point clearer. Is RP used in the current experiments? Then how does this influence the results? Is the RP used for computational purposes? Then can we quantify the gain in computation?
> >
> > R: Generally speaking, representation learning for big datasets is usually costly in storage and computation. For example, the dimension of images in STL-10 is 96 \times 96 \times 3 (i.e., the vector length is 27648). This makes it almost impossible to directly estimate the gradient of MI between the input and representation. Therefore, we introduce Random Projection to make MIGE applicable high dimensional settings. For exmaple on STL-10., when the dimension of images is reduced to 256 via RP, we can observe significant improvement over the baselines (as shown in the table below). More specific analyses will be added to the revised manuscript.
> >
> > Table: Classification accuracy (top 1) results on STL-10.
> > RP denotes Random Projection.
> > .----------------------------------------------------------------
> > .							STL-10
> > .					conv	fc		Y
> > .----------------------------------------------------------------
> > .DIM(JSD)			42.03	30.28	28.09
> > .DIM(infoNCE)		43.13	35.80	34.44
> > .----------------------------------------------------------------
> > .MIGE			unaffordable computational cost
> > .MIGE+RP to 1024d	49.08	40.09	38.95
> > .MIGE+RP to 512d	49.89	41.05	38.56
> > .MIGE+RP to 256d	49.91	40.24	38.83
> > .----------------------------------------------------------------
> >
> > Q5: In practice, we do not care about MI estimation’. Please explain further or refer to previous work. *
> >
> > R:Sorry for the confusion. Existing deep models focusing on mutual information usually firstly estimate the MI, and then maximize/minimize it by gradient ascent/descent. Our concern is that directly estimating the gradient could be easier and more accurate since estimating the mutual information in itself is very difficult. We will revise this sentence.
> >
> > Q6: Two points 1. The information ‘between z and z’ is probably a typo? 2. How does sufficiency relate to an optimization problem? Doesn’t sufficiency mean in this context I(X;Y)=I(Z;Y)?
> >
> > R:Yes, it is a typo. Here sufficiency means this context I(X;Y)=I(Z;Y).
> >
> > Q7: Equation 12: In the part . Why do we take gradient w.r.t. x? It seems to me that the reparametrization is a function of x only via. If not, then please explain what this tuple means.
> >
> > R: While x is unrelated to parameter \psi, we cannot directly get the gradient of log p_\psi(x,z) with z. p_\psi(x,z) is an implicit distribution. Hence, we don’t know the formation of p_\psi(x,z). But based on Spectral Stein Gradient Estimator, we can estimate the gradient of log p_\psi(x,z) w.r.t (x,z) by sampling (x,z). Then the gradient of log p_\psi(x,z) w.r.t \psi is directly computed according to the chain rule derivation.
> >
> > Q8: Table 1 has no units. How to interpret the numbers in this table?
> >
> > R: Table 1 shows the classification accuracy (top 1) results of downstream tasks on CIFAR10 and CIFAR100 in Deep InfoMax experiments. "%" will be added.
> >
> > Q9: Section 4.3, authors note their experiment is ‘a little bit different’ from other related research. How and what exactly is different?
> > R: We use the same setting except that the initial learning rate of 2e-4 was set  for Adam optimizer, and exponential decay with decaying rate by a factor of 0.96 was set for every 2 epochs.
> >
> > [1] Ishmael Belghazi, Aristide Baratin, Sai Rajeswar, Sherjil Ozair, Yoshua Bengio, Aaron Courville, and R Devon Hjelm. Mine: mutual information neural estimation. arXiv preprint arXiv:1801.04062, ICML, 2018.
> > [2] R Devon Hjelm, Alex Fedorov, Samuel Lavoie-Marchildon, Karan Grewal, Phil Bachman, Adam Trischler, and Yoshua Bengio. Learning deep representations by mutual information estimation and maximization. In International Conference on Learning Representations, 2019.
> > [3] Ben Poole, Sherjil Ozair, Aaron van den Oord, Alex Alemi, and George Tucker. On variational bounds of mutual information. In International Conference on Machine Learning, 2019.

---

> > > ### Comment · AnonReviewer3 · 2019-11-14
> > > **Response to authors**
> > >
> > > Thank you for responding
> > >
> > > >> On the copy-editor. I don’t think hiring a copy editor is necessary. I recommend some off-the-shelf grammar and spelling checkers. Also, having the paper proofread by colleagues generally helps. They would point out missing definitions of variables, like ‘AnonReviewer2’ pointed out.
> > >
> > > >> Q1Q2
> > > Even some empirical statistics would suffice. I find it misleading to plot only one realization of the gradient estimate. How about plotting 10000 realizations and calculate the variance per value of \rho? If that number is lower than MINE, it would strengthen your argument.
> > >
> > > >> Q3
> > > Ok. Then it’s a fair comparison. If you used their code in that way, please consider citing it as such.
> > >
> > > >> Q4
> > > Ok. Do you plan to include that table in the revision?
> > >
> > > >> Q5
> > > In other comments in this thread you mentioned the concurrent work [1]. How does [1] relate to the statement you make in the paper?
> > >
> > > >> Q6Q8Q9
> > > Understood. Please include in the revised version
> > >
> > > [1] https://openreview.net/forum?id=rkxoh24FPH

---

> > > > ### Author Response · Authors · 2019-11-15
> > > > **Response to Reviewer 3**
> > > >
> > > > Thank you for your response
> > > >
> > > > >> Q1Q2
> > > > Even some empirical statistics would suffice. I find it misleading to plot only one realization of the gradient estimate. How about plotting 10000 realizations and calculate the variance per value of \rho? If that number is lower than MINE, it would strengthen your argument.
> > > >
> > > > R：Thank you for your constructive suggestion.We agree that the method of  quantitive measures of  proposed by reviewer  is effective to strengthen our argument. Due to the limited response time, we will add the quantitive measures for tightness and variance of MIGE  in the camera-ready.
> > > >
> > > >
> > > > >> Q3
> > > > Ok. Then it’s a fair comparison. If you used their code in that way, please consider citing it as such.
> > > >
> > > > R：Yes, of course.
> > > >
> > > > >> Q4
> > > > Ok. Do you plan to include that table in the revision?
> > > > Yes, of course.
> > > >
> > > > >> Q5
> > > > In other comments in this thread you mentioned the concurrent work [1]. How does [1] relate to the statement you make in the paper?
> > > > We will add the statement related to [1] to the appendix
> > > >
> > > > >> Q6Q8Q9
> > > > Understood. Please include in the revised version
> > > >
> > > > R：Yes, of course.
> > > >
> > > > [1] https://openreview.net/forum?id=rkxoh24FPH

---

### Official Review · AnonReviewer2 · 2019-10-31
**Official Blind Review #2**

**Rating:** 3

**Review:**

The paper argues that directly estimating the intractable mutual information (MI) for representation learning is challenging in high dimensions. Instead the authors propose to estimate the needed MI gradients directly using a score function based approach. Using some identities of MI the authors arrive at an expression for the gradient of the mutual information between input and latent representation (eq 10) and proposes to use a generalization of the reparameterization trick and spectral stein gradient descent to approximate this gradient. In Toy experiment and MNIST/CIFAR10 experiments the authors demonstrate that their method produces latent representations that are more informative than competing MI methods for downstream classification tasks. I found the approach and content of the paper interesting and the results seems encouraging.  My main concern is that I did not find the that I did not find the method and experimental section to be fully comprehensive and further lacking many details which makes it hard to compare the results with prior work.


Pros:
1) I find the approach taken by the authors interesting and different from current MI estimation approaches. The paper convincingly motivates their approach by describing the deficiencies of current MI estimators and why targeting the gradients directly might have merits.
2) The authors propose to use SSGD and 'generalized' reparameterization in a (well motivated) new setting.
3) The cifar10 experiments in table 1 are encouraging and the toy experiment in 2D is illustrates nicely the deficiencies of the current MI estimators

Cons
1) The experimental section is lacking many details to fully understand how and what experiments were performed and how comparable they are to prior work
 2) The paper would benefit greatly from a thorough editing to clarify the presentation - there are many missing concepts and definitions that makes it hard to follow without intimate knowledge of related literature.


Further suggestions / questions

1. In section 3. Please define q(z)_psi(z),  q(x,z)_psi and describe how they relate to  E_psi.

2) What exactly are the contributions by the authors wrt to spectral stein gradient descent (sec 2.1) e.g. is it the scalable approach based on random projections described in sec 3 ? Further i would like some discussion on the quality of this approximation?

3) Please provide some more details on the DeepInfoMax and Information bottleneck experiments e.g. How exactly did you estimate the MI gradients in these settings? how is the downstream task setup and is it identical to prior work?


4) About writing style:
I think it would benefit the paper if you let the reader decide for them self what adjectives should be used to describe a result. A few concrete suggestions:
 - Use remarkable/y about your own findings a bit more sparingly (used 4x).
 - Consider deleting “much” and “vast” in a sentence like: “our approach MIGE gives much more favorable gradient direction, and demonstrates more power in controlling information flows without vast loss”.

**Experience Assessment:**

I have read many papers in this area.

**Review Assessment: Checking Correctness Of Derivations And Theory:**

I assessed the sensibility of the derivations and theory.

**Review Assessment: Checking Correctness Of Experiments:**

I assessed the sensibility of the experiments.

**Review Assessment: Thoroughness In Paper Reading:**

I read the paper at least twice and used my best judgement in assessing the paper.

---

> ### Author Response · Authors · 2019-11-13
> **Response to Reviewer 2**
>
> Thank you for acknowledging that our idea is interesting and the results are encouraging. We will revise our paper according to the comments and hire a copy-editor to carefully polish the writing.
>
> In the following, we will answer the concerns of the reviewer.
>
> 1) For experiments, we follow the experiments of Deep InfoMax and Information Bottleneck to set the experimental setup as in [1, 2], and we also refer to their source code [3, 4]. Under these experimental settings, we use our MI Gradient Estimator to replace the MI estimator in Deep InfoMax and Information Bottleneck.
> We will provide an algorithm description in the revision, and we will release our code upon acceptance with more detailed settings.
>
> 2) Due to the page limits (recommended 8 pages in ICLR CALL for Papers), we cannot provide more details of missing concepts and definitions. We will add an appendix and make our paper more self-contained in the revision.
>
> Q1: In section 3. Please define q(z)_psi(z), q(x,z)_psi and describe how they relate to E_psi.
>
> R: For definitions of q(z)_\psi(z) and q(x,z)_\psi, q(z)_\psi(z) corresponds to the distribution of representation of x via the encoder E_\psi as described in Equation (11). As mentioned in our paper, "we can obtain the samples from the marginal distribution of z by pushing samples from the data empirical distribution p(x) through E\psi(.) for representation learning." q_\psi is an implicit distribution determined by the encoder parameters \psi. q(x,z)_\psi is the joint distribution of (x,z), which is determined by the encoder parameters \psi.
>
> Q2: What exactly are the contributions by the authors wrt to spectral stein gradient descent (sec 2.1) e.g. is it the scalable approach based on random projections described in sec 3? Further i would like some discussion on the quality of this approximation?
>
> R: Generally, representation learning for big datasets is usually costly in storage and computation. For example, the dimension of images in STL-10 is 96 \times 96 \times 3 (the dimension is 27648). This makes it hard to directly estimate the gradient of MI between the input and representation by SSGE. Therefore, we introduce Random Projection to achieve the approximation of Equation (19) in the Nyström approximations of SSGE.
> Thus with the aid of Random Projection, we could evaluate on bigger datasets, e.g., STL-10. Especially, when the dimension of images is reduced to 256, we can observe significant improvement over the baselines (as shown in the table below). More specific analyses will be added to the revised manuscript.
>
> Table: Classification accuracy (top 1) results on STL-10.
> RP denotes Random Projection.
> .----------------------------------------------------------------
> .							STL-10
> .					conv	fc		Y
> .----------------------------------------------------------------
> .DIM(JSD)			42.03	30.28	28.09
> .DIM(infoNCE)		43.13	35.80	34.44
> .----------------------------------------------------------------
> .MIGE			unaffordable computational cost
> .MIGE+RP to 1024d	49.08	40.09	38.95
> .MIGE+RP to 512d	49.89	41.05	38.56
> .MIGE+RP to 256d	49.91	40.24	38.83
> .----------------------------------------------------------------
>
> Q3: Please provide some more details on the DeepInfoMax and Information bottleneck experiments e.g. How exactly did you estimate the MI gradients in these settings? how is the downstream task setup and is it identical to prior work?
>
> R: We follow the experiments of Deep InfoMax and Information Bottleneck to set the experimental setup as in [1, 2], and we also refer to their source code [3, 4]. Under these experimental settings, we use our MI Gradient Estimator to replace the MI estimator in Deep InfoMax and Information Bottleneck.
> We will provide an algorithm description in the revision, and we will release our code upon acceptance with more detailed settings.
>
> Q4: About writing style
>
> R: Thank you for your valuable suggestions about writing style, we will fix these problems in the revision.
>
> [1] R Devon Hjelm, Alex Fedorov, Samuel Lavoie-Marchildon, Karan Grewal, Phil Bachman, Adam Trischler, and Yoshua Bengio. Learning deep representations by mutual information estimation and maximization. In International Conference on Learning Representations, 2019.
> [2] Alemi, A. A., Fischer, I., Dillon, J. V., and Murphy, K. Deep variational information bottleneck. arXiv preprint arXiv:1612.00410, 2016.
> [3] https://github.com/rdevon/DIM
> [4] https://github.com/alexalemi/vib_demo

---

> > ### Comment · AnonReviewer2 · 2019-11-14
> > **Response to Authors**
> >
> > "For experiments, we follow the experiments of Deep InfoMax and Information Bottleneck to set the experimental setup as in [1, 2], and we also refer to their source code [3, 4]. Under these experimental settings, we use our MI Gradient Estimator to replace the MI estimator in Deep InfoMax and Information Bottleneck"
> >
> > Thanks - Please clarify this in the main text. Also i just looked over the PDF again and i could not find a link to the code you refer to?
> >
> > "For definitions of q(z)_\psi(z) and q(x,z)_\psi, q(z)_\psi(z) corresponds to the distribution of representation of x via the encoder E_\psi as described in Equation (11). As mentioned in our paper, "we can obtain the samples from the marginal distribution of z by pushing samples from the data empirical distribution p(x) through E\psi(.) for representation learning." q_\psi is an implicit distribution determined by the encoder parameters \psi. q(x,z)_\psi is the joint distribution of (x,z), which is determined by the encoder parameters \psi."
> >
> > Again thanks for the clarification - I would strongly encourage you to include this in the main text as well as it is not clear how E_psi, q_psi etc is related.
> >
> > Thanks for providing the STL results - Can you give any intuition for why you see decreased performance when increasing the dimensionality of RP?

---

> > > ### Author Response · Authors · 2019-11-14
> > > **Response to Reviewer 2**
> > >
> > > R1 : We will revise our paper to include your valuable comments.
> > >
> > > R2: Note the performace of Random Project in different layers is different. And the perfomace of the last layer is quite invariant with different RP dimensons. We will   conduct more extensive expreiments with different dimensions, and will show the trend curve in the camera-ready upon acceptance.

---

### Official Review · AnonReviewer4 · 2019-11-01
**Official Blind Review #4**

**Rating:** 6

**Review:**

This paper proposes the Mutual Information Gradient Estimator (MIGE) for estimating the gradient of the mutual information (MI) instead of calculating it directly in learning representation. They are using Stein's estimator following by a random projection to build a tractable approximation to the gradient of the MI.
 The MIGE is evaluated on several of unsupervised and supervised tasks, and shown improvement over prior MI estimation approaches in maximize the MI and learning features for classification.

In general, I think that the idea of estimating the gradient of the MI instead of directly calculating it  is an exciting research direction, and this paper combines a few pieces together (As mentioned in the paper, there was a work of Li & Turner, 2017 that applied Stein's estimator for implicit models).
However, the experimental part of this paper is lacking. My main concern is regarding the performance on downstream tasks. Although the experiments demonstrate wins over different models in maximizing MI for CIFAR10 and CIFAR100, the only comparison for downstream tasks is for Permutation-invariant MNIST. One more concern is regarding the random projection. It is not clear what is the effect of it on the representation, and how it impacts on the gradine's estimation.

Strengths:
+ Interesting new model for representation learning based on an estimation of the MI gradients'.
+ Good set experiments looking at MI maximization performance.
+A well-written and well-organized paper.

Weaknesses:
 No comparison on downstream tasks for more datasets except MNIST. In the end, a key question is a final accuracy on different datasets and how to maximize the information effect on it.
There is no discussion about the effect of the random projection on the representation. For example, how it affects performance? How much the algorithm sensitive to this projection? What is the performance of the MINE if it combined with random projection...

Minor comments:

-Typos and English mistakes - there are many typos. For example -
    In the introduction - "Another closely related work is the the Information…"
    In section 2 - “In order to overcome this disadvantages"
    In section  2.20  - "In optimization, it should be achieved by maximizing the information between z and z."
- There should be more detailed explanations of the experiments. For example - what is the projected dimension (for all the experiments).


**Experience Assessment:**

I have published in this field for several years.

**Review Assessment: Checking Correctness Of Derivations And Theory:**

I assessed the sensibility of the derivations and theory.

**Review Assessment: Checking Correctness Of Experiments:**

I assessed the sensibility of the experiments.

**Review Assessment: Thoroughness In Paper Reading:**

I read the paper thoroughly.

---

> ### Author Response · Authors · 2019-11-13
> **Response to Reviewer 4**
>
>
> Thank you for acknowledging that our idea is interesting and the results are encouraging. We will revise our paper according to the comments and hire a copy-editor to carefully polish the writing.
>
> Q1: No comparison on downstream tasks for more datasets except MNIST. In the end, a key question is a final accuracy on different datasets and how to maximize the information effect on it.
> R: Table 1 shows the classification accuracy (top 1) results of downstream tasks on CIFAR10 and CIFAR100 in Deep InfoMax experiments. Table 2 shows the experimental results on MNIST in Information Bottleneck (IB).
>
> Q2: There is no discussion about the effect of the random projection on the representation. For example, how it affects performance? How much the algorithm sensitive to this projection?
> R: Generally speaking, representation learning for big datasets is usually costly in storage and computation. For example, the dimension of images in STL-10 is 96 \times 96 \times 3 (i.e., the vector length is 27648). This makes it almost impossible to directly estimate the gradient of MI between the input and representation. Therefore, we introduce Random Projection to make MIGE applicable high dimensional settings. For exmaple on STL-10., when the dimension of images is reduced to 256 via RP, we can observe significant improvement over the baselines (as shown in the table below). More specific analyses will be added to the revised manuscript.
>
>
> Table: Classification accuracy (top 1) results on STL-10.
> RP denotes Random Projection.
> .----------------------------------------------------------------
> .							STL-10
> .					conv	fc		Y
> .----------------------------------------------------------------
> .DIM(JSD)			42.03	30.28	28.09
> .DIM(infoNCE)		43.13	35.80	34.44
> .----------------------------------------------------------------
> .MIGE			unaffordable computational cost
> .MIGE+RP to 1024d	49.08	40.09	38.95
> .MIGE+RP to 512d	49.89	41.05	38.56
> .MIGE+RP to 256d	49.91	40.24	38.83
> .----------------------------------------------------------------
>
> Q3: What is the performance of the MINE if it combined with random projection
>
> R: It is possible to apply RP for MINE.  However,  due to the dependence on the discriminator used to estimate the mutual information,  RP+MINE may lead to results with high variance. And we will investigate this issue in the future work.

---

### Decision · Program_Chairs · 2019-12-19

**Decision:**

Accept (Poster)

**Comment:**

This paper proposes the Mutual Information Gradient Estimator (MIGE) for estimating the gradient of the mutual information (MI), instead of calculating it directly. To build a tractable approximation to the gradient of MI, the authors make use of Stein's estimator followed by a random projection. The authors empirically evaluate the performance on representation learning tasks and show benefits over prior MI estimation methods.
The reviewers agree that the problem is important and challenging, and that the proposed approach is novel and principled. While there were some concerns about the empirical evaluation, most of the issues were addressed during the discussion phase. I will hence recommend acceptance of this paper. We ask the authors to update the manuscript as discussed.